

# Neural Network Model for Automated Prediction of Avalanche Danger Level

Vipasana Sharma[1], Sushil Kumar[2], Rama Sushil[3]

[1] Student Intern, Birla Institute of Technology & Sciences, Pilani, Hyderabad Campus, India
[2] Ex-Scientist, Wadia Institute of Himalayan Geology, Uttarakhand, India
[3] Professor of Computing, DIT University, Uttarakhand, India

*Correspondence:* Vipasana Sharma (vipasanasharma78@gmail.com)

**Abstract** Snow avalanches cause danger to human lives and property worldwide in high-altitude mountainous regions. Mathematical models based on past data records can predict the danger level. In this paper, we are proposing a neural network

model for predicting avalanches. The model is trained with a quality-controlled sub-dataset of Swiss Alps. Training accuracy of 79.75% and validation accuracy of 76.54% have been achieved. Comparative analysis of neural network and random forest models concerning metrics like precision, recall, and F1 has also been carried out.

## 1. Introduction

Accurate prediction of snow avalanches can help ensure people's safety in snow-covered regions. Many countries still depend
on human experts to analyse meteorological data to forecast avalanche warnings.

The major hurdle in developing machine learning models is the lack of sufficient and reliable data. This issue has been resolved to a great extent by the WSL Institute of Snow and Avalanche Research, Switzerland, by collecting 20 years of data in avalanche forecasting. This data set has been further refined with quality control by experts. The dataset combines different feature sets with meteorological variables.

This unique dataset has enabled experimentation with machine learning models like neural networks and compared its performance with the random forest machine learning technique.

This paper is organized as follows. Related literature is briefly overviewed in Section II. The dataset used for the training of neural networks is described in Section III. After that, in Section IV, we explain the neural network model, tuning of hyper-parameters, and evaluation metrics. Random Forest machine learning method details applied to the same dataset are described

in Section V. Results from both methods are compared and analysed in Section VI. The paper is concluded in Section VII.

## 2. Related Work

Many countries face snow avalanche hazards with snow-clad mountains. It affects people, facilities, and properties. The impact of snow avalanches on living, work, and recreation in Canada is well documented (Sethem et. al., 2003). Every country



generally follows its own avalanche classification system. However, in this work, we will follow the European Avalanche

Danger Scale (EAWS, 2018).

A comprehensive dataset with the meteorological variables (resampled 24-hour averages) and the profile variables extracted from the simulated profiles has been created (Pérez-Guillén et al., 2022). Weather station data of the IMIS network in Switzerland for dry-snow conditions are further quality controlled for creating 29,296 records. Each record has 30 variables. The benefits and challenges of using machine learning and AI for avalanche forecasting in Norway and Canada have been

discussed in detail (Horton S. et al., 2020). Also, machine learning algorithms like the random forest has been successfully used for the prediction of snow avalanches in the region of the Swiss alps (Pérez-Guillén et al., 2022). The random forest technique has also been used for forecasting snow avalanches in the Himalayan region (Chawla. M. et al., 2021).

### 3. Dataset

In this paper, the public data set provided by Envidat, a Swiss organization, is used. This data is verified and supported by the

Swiss Data Science Centre (Grant/Award: grant C18-05 "DEEP snow"). More than 20 years of data for avalanche forecasting in the Swiss Alps is provided. Data covers the Swiss winters from 1997-2017. The data is collected from 182 snow stations and is used by the Swiss avalanche warning service.

The data set includes the meteorological variables (resampled 24-hour averages) and the profile variables extracted from the simulated profiles. The data set contains the danger ratings published in the official Swiss avalanche bulletin using

SNOWPACK simulations. The SNOWPACK simulations provide two different output files for each station: (i) time series of meteorological variables and (ii) simulated snow cover profiles.

This study uses measured, extracted, profiled, and modelled variables. Thus, 30 variables are shown in Table 1 for training neural network models for predicting snow avalanches.

### 4. Proposed Neural Network: NNM-1

Neural network models allow the modelling of complex nonlinear relationships between the multiple input and output variables. It is a network of input, output, and intermediate layers (Figure 1).). The outputs are obtained by a linear combination of the weights with inputs. The weights are selected using a "learning algorithm" that minimizes a "cost function."

This study uses multilayer feed-forward networks, where each layer of nodes receives inputs from the previous layers. The outputs of the nodes in one layer are inputs to the next layer. For example, the inputs into the hidden neuron in Figure 1 are

combined linearly to give the following output.

$$z_j = b_j + \sum w_{ij} . x_j \qquad (1)$$

where $z_j$ denotes the hypothesis of parameters w and b, $x_j$ denotes the features in the training set.




A nonlinear function modifies the above outputs of nodes before being used as inputs by the next layer. The parameters $b_j$ and

$w_{ij}$ are learned from data. The number of hidden layers and nodes in each hidden layer are specified in advance.

Training of artificial neural networks, also known as supervised learning, involves adjusting weights until the model is properly

fitted with Labels indicating the avalanche danger according to European norms. A total of 30 input variables are used for

training the network. The avalanche threat is categorized into five zones as follows:

● 1 – Low

   ● 2 – Moderate

   ● 3 – Considerate

   ● 4 –High

   ● 5– Very High

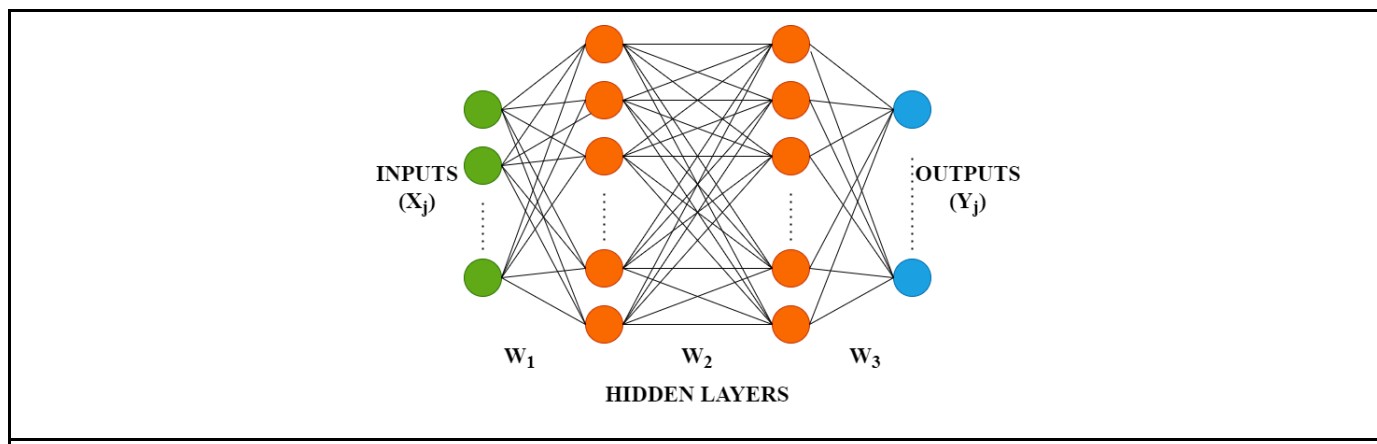

**Figure 1: Multilayer Neural Network model**

The model's performance across the training dataset is described by a Loss function which computes the difference between

the trained model's predictions and the actual incident instances. The loss function would be very high if the gap between

expected and actual results is too large. The loss function gradually learns to lower the prediction error with optimization

function (Bottou, 1991). A multi-class classification cost function is used for avalanche prediction for each danger level.

The average difference between the probability distributions that were anticipated and that occurred is calculated.





**Table 1 Meteorological thirty variables in four categories (Measured, Extracted, Profiled, Modelled) (Pérez-Guillén et al., 2022) used to develop the neural network model.**

| (a) Measured variables | (b) Extracted variables | (c) Profiled variables |
|---|---|---|
| Air temperature | 3 d wind drift | Min critical cut length at a deeper layer of the penetration depth |
| Wind velocity | 7 d wind drift | Critical cut length at surface weak layer |
| Relative humidity | 7 d sum of daily height of new snow | Natural stability index at surface layer |
| Wind velocity drift | | Skier penetration depth |
| (d) Modelled variable | | |
| Sensible heat | Sk38 skier stability index | Surface temperature |
| Ground heat at soil interface | Diffuse incoming shortwave | Solid precipitation rate |
| Incoming long-wave radiation | Depth of Sk38 skier stability index | Snow height |
| Net long-wave radiation | Natural stability index | 24 h height of new snow |
| Incoming shortwave radiation | Depth of natural stability index | 3 d sum of daily height of new snow |
| Net shortwave radiation | Structural stability index | 24 h wind drift |
| Parameterized albedo | | |


$$Cross\ Entropy\ loss = -\sum_{i=1}^{5} y_i \cdot log\ \hat{y}_i \quad (2)$$

Equation (2) computes cross entropy loss using the target and predicted danger levels..

In our scenario, the output layer is set up with five nodes (one for each danger level). "SoftMax" activation function is used to compute the probability for each danger class $z_j$


$$Softmax(z_j) = \frac{e^{z_i}}{\sum_j e^{z_j}} \quad\quad (3)$$

Equation (3) transforms the raw outputs of the neural network into probabilities (Christopher, 2005).



The gradient descent method has been used to update the weights and bias through backpropagation. The "Adam" (Adaptive moment Estimation) optimizer is used for optimization. It performs the search process using an exponentially decreasing

moving average of the gradient.

The performance of a neural network mainly depends on the number of hidden layers and the number of neurons in the respective hidden layer. Table 2 shows the range of hyperparameters used for testing different neural networks. The upper limit on the number of neurons has been set according to the number of raw variables in the original data set.

**Table 2: Range of hyper-parameters used for testing neural networks**

| Hyper-parameter | Minimum value | Maximum value |
|---|---|---|
| Number of hidden layers | 1 | 10 |
| Number of neurons in the hidden layer | 5 | 68 |


After exhaustive testing of neural networks with hyper-parameters (Table 2), networks are ranked according to training accuracy (Table 3). However, after plotting of training and validation curves (Figure 2 a, c, e, g), it is observed that validation accuracy was reduced through training accuracy was increased. Thus, indicating over-fitting of the neural network models. To address this problem, dropout layers are included and tested with different dropouts. Figure 2 b, d, f, h shows that dropout

regularization successfully resolved over-fitting and significantly improved validation accuracy. Maximum validation accuracy is observed for NNM#3 with a dropout of 0.2, 0.1, and 0.1 on three hidden layers (Figure 3).

**Table 3:** Neural network models ranked according to training accuracy with a learning rate of 0.001 for 100 epochs and batch size of 64

| | Number of hidden layers | Number of nodes in the first layer | Number of nodes in remaining layers | Training accuracy (%) | Validation accuracy (%) |
|---|---|---|---|---|---|
| NNM#1 | 3 | 50 | 25,16 | 84.90 | 72.15 |
| NNM#2 | 3 | 48 | 24,16 | 79.19 | 74.11 |
| NNM#3 | 2 | 48 | 24 | 79.09 | 73.06 |
| NNM#4 | 3 | 36 | 24,16 | 78.70 | 72.64 |







**Figure 2** Effect of Dropout on training and validation accuracy (a) NNM#1 without Dropout (b) NNM#1 with a dropout of 0.2, 0.1, and 0.1 on three hidden layers (c) NNM#2 without Dropout (d) NNM#2 with a dropout of 0.2, 0.1 and 0.1 on three hidden layers (e) NNM#3 without Dropout (f) NNM#3 with a dropout of 0.2 and 0.1 on two hidden layers (g) NNM#4 without Dropout (h) NNM#4 with a dropout of 0.1, 0.1 and 0.1 on three hidden layers


The proposed neural network architecture based on the above study is shown in Table 4. It has three hidden layers and has been trained for 100 epochs. The model achieved a Training Accuracy of 79.75% and a Validation Accuracy of 76.54%. A confusion matrix for the proposed NNM-1 is shown in Figure 5, predicted a considerate danger level; out of 1000 cases, 806
cases of true positive and 194 cases of false positive.





**Table 4: Proposed neural network model (NNM-1) architecture**

| Number of inputs | Number of hidden layers | Number of nodes in layers | Learning rate | Epoch | Batch size | Dropout | Number of outputs |
|---|---|---|---|---|---|---|---|
| 30 | 3 | 48,24,16 | 0.001 | 100 | 64 | 0.2, 0.1, 0.1 | 5 |


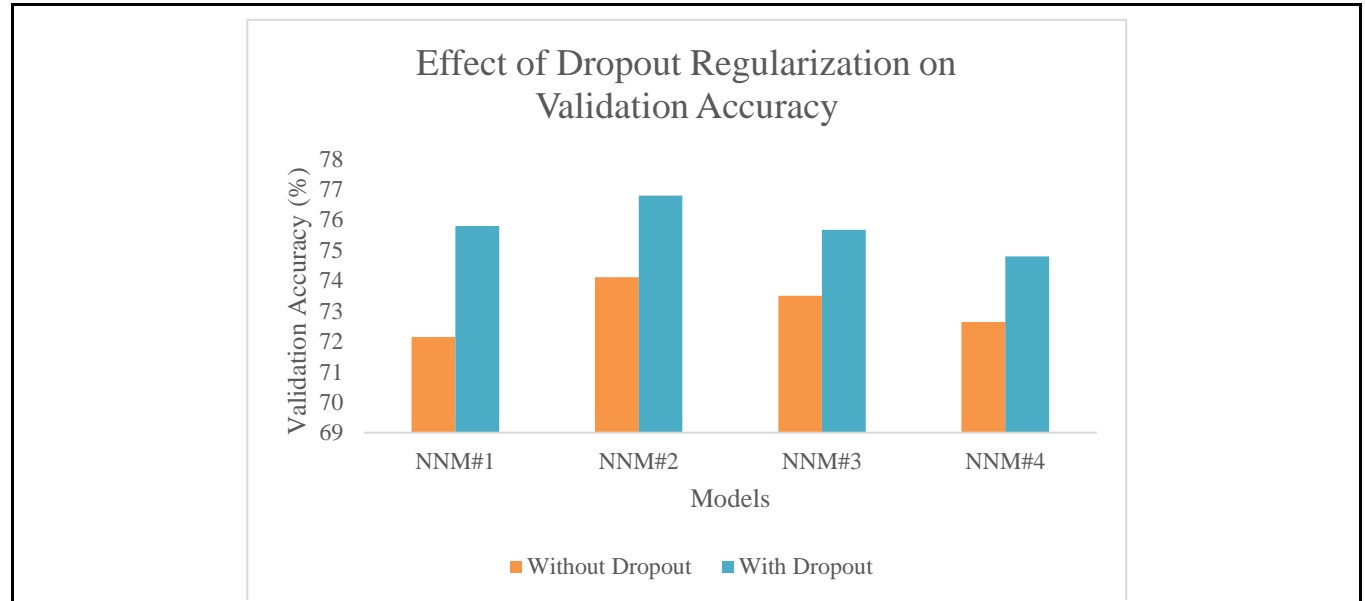

**Figure 3**: Improvement in validation accuracy of neural network models with Dropout regularization. Maximum validation accuracy was achieved with NNM#2 with Dropout of 0.2, 0.1, and 0.1 on three hidden layers

## 4.1 Evaluation Matrix for NNM-1:

Table 4 shows the various evaluation metrics like Accuracy, Precision, Recall, and F1 Score for the Neural Network model. The proposed neural network correctly predicted 76 classifications for every 100 forecasts made. The macro and weighted averages of precision, recall, and F1 Score are shown in Table 4. Macro average is computed without considering the proportion of labels in different classes of danger levels. It may be noted that weighted average takes into account low number of labels for high and very high danger level classes. The proposed neural network model achieved macro and weighted average

F1 Score of 0.69 and 0.75, respectively.




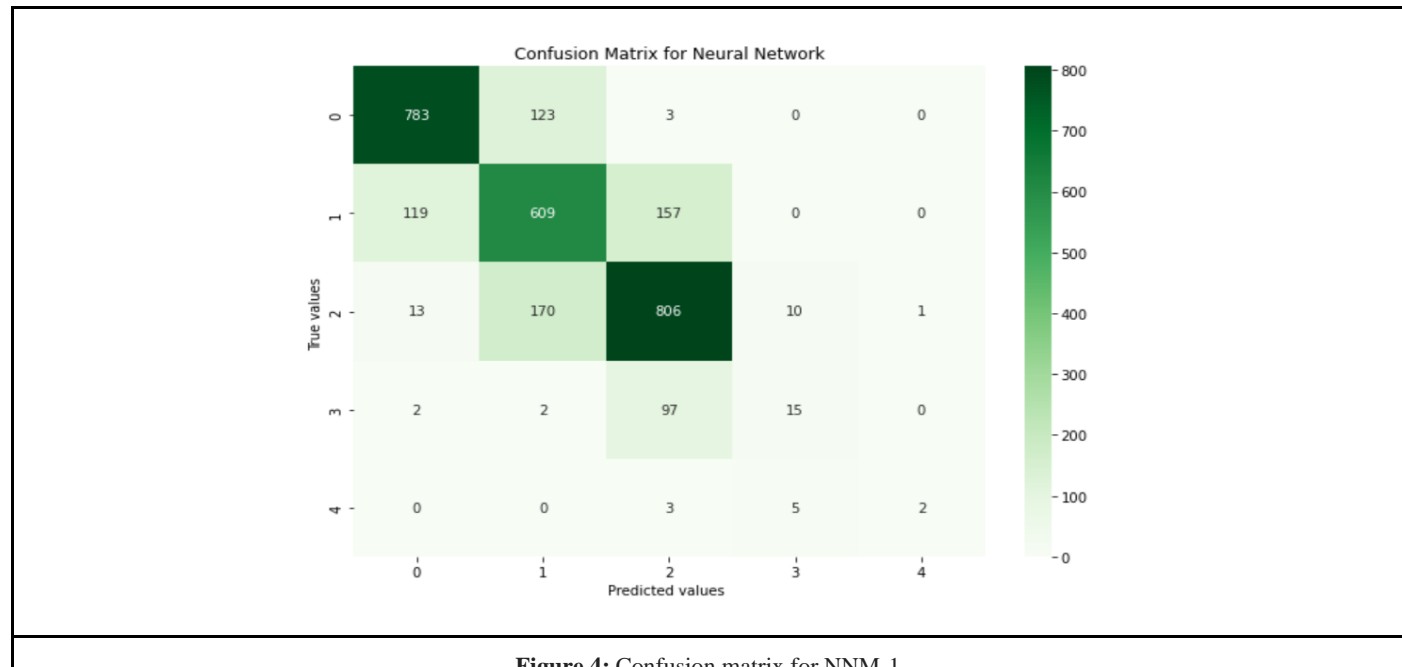

**Figure 4:** Confusion matrix for NNM-1


**Table 4: Parametric evaluation metrics for the proposed Neural Network Model**

| Class | Danger level | Precision | Recall | F1 | Support |
|---|---|---|---|---|---|
| 0 | LOW | 0.85 | 0.86 | 0.86 | 909 |
| 1 | MODERATE | 0.67 | 0.69 | 0.68 | 885 |
| 2 | CONSIDERATE | 0.76 | 0.81 | 0.78 | 1000 |
| 3 | HIGH | 0.50 | 0.13 | 0.21 | 116 |
| 4 | VERY HIGH | 0.67 | 0.20 | 0.31 | 10 |
| | **Accuracy =0.76** | | | | 2920 |
| | **MACRO AVG** | 0.69 | 0.54 | 0.57 | 2920 |



| | | | | |
|---|---|---|---|---|
| **WEIGHTED AVG** | 0.75 | 0.76 | 0.75 | 2920 |

# 5. Random Forest

A random forest is a meta estimator that employs averaging to increase predictive accuracy and reduce over-fitting after fitting numerous decision tree classifiers to different dataset subsamples. A subset of the training data is randomly chosen by the

Random Forest classifier to construct a set of decision trees. It simply consists of a collection of decision trees (DT) from a randomly chosen subset of the training set, which is subsequently used to decide the final prediction. The confusion matrix for the Random Forest classifier is shown in Table 5. The data set (2920 records) used for validating the Neural Network model is applied to the computing performance matrix. Several decision trees make up the random forest model, which is trained with the Classification and Regression Tree (CART) algorithm. Table 5 shows the various evaluation metrics like precision,

Recall, and F1 Score for the Random Forest model.

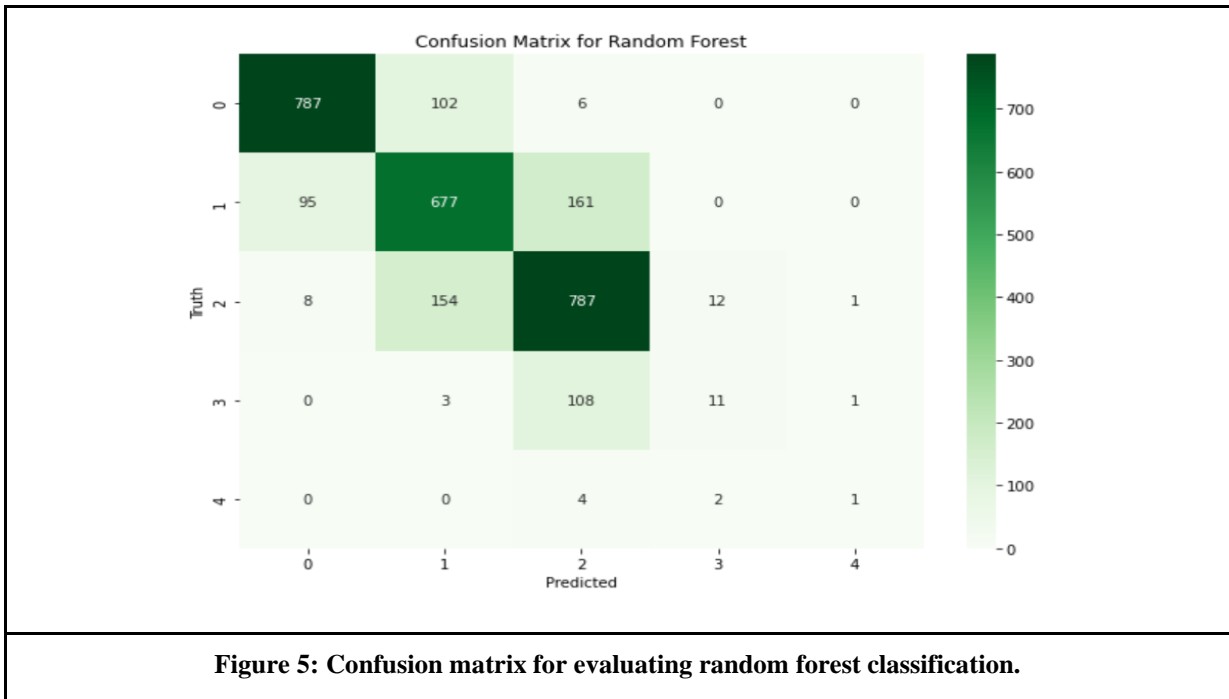

**Figure 5: Confusion matrix for evaluating random forest classification.**

# 6. Results and discussion





Testing of the proposed model has been carried out with 2920 records for which ground truth labels are available. The data for
the high and very high avalanche threats is less compared to low, moderate, and considerate threats. This scenario in a data set
where samples of data in one class are much higher compared to that of the other class is a skewed data set. In this case, the
higher data sample class (Low, moderate, and considerate avalanche threat) becomes the major class. The class consisting of
relatively fewer data samples (high and very high avalanche threat) is labelled as a minor class. Hence, the overall neural
network performance is affected, thereby generating less accurate results for the minor class.


| **Table 5:** Evaluation for the Random Forest model (RF-A) | | | | | |
|---|---|---|---|---|---|
| **Class** | **Danger level** | **Precision** | **Recall** | **F1** | **Support** |
| 0 | Low | 0.88 | 0.88 | 0.88 | 895 |
| 1 | Moderate | 0.72 | 0.73 | 0.72 | 933 |
| 2 | Considerate | 0.74 | 0.82 | 0.78 | 962 |
| 3 | High | 0.44 | 0.09 | 0.15 | 123 |
| 4 | Very High | 0.33 | 0.14 | 0.20 | 7 |
| | **Accuracy =**0.76 | | | | 2920 |
| **Macro Avg** | | 0.69 | 0.54 | 0.57 | 2920 |
| **Weighted Avg** | | 0.75 | 0.76 | 0.75 | 2920 |

We trained multiple neural network models with a variety of hyper-parameters. The model NNM-1 (Table 4) used for the
comparative analysis is without over-fitting and has maximum validation accuracy. Another Random Forest model RF-A
(Möhle et al., 2014) is also tested with the same data sets. Both models (NNM-1 and RF-A) achieved the same overall accuracy
(0.76) as RF-1, which is slightly less than RF-2 accuracy (0.78). F1 scores for Low, Medium and Considerate classes are equal
for NNM-1 and RF-A models (Table 6). However, low F1 value for High and Very High class for NNM-1 and RF-A is
attributed to skewed data distribution. Weighted average values (Table 5) are more appropriate as compared to macro average
as these consider low number of labels for class 3 and class 4.




## 7. Conclusion

A neural network model to predict avalanche danger levels has been developed. The model is validated by using 20 years of meteorological measurements and extracted and modelled variables of the Swiss Alps. Extensive testing has been carried out for tuning hyperparameters, like the number of hidden layers and neurons. The data used for testing the neural network model

is also applied to the random forest model for the evaluation of performance metrics. The developed model has achieved a training accuracy of 79.75% and a validation accuracy of 76.54% which is same as of RF-1 and RF-A but 2.56 % less accuracy than RF-2.

| Table 6: Various parametric values of some existing models and proposed NNM-1, for snow avalanche prediction. |||||||||||
|---|---|---|---|---|---|---|---|---|---|---|
| **Model** | **DL** | **Prec.** | **Recall** | **F1** | **Support** | **Model** | **Prec.** | **Recall** | **F1** | **Support** |
| (a) | Low | 0.85 | 0.86 | 0.86 | 909 | (b) | 0.88 | 0.88 | 0.88 | 895 |
| NNM-1 | Medium | 0.67 | 0.69 | 0.68 | 885 | RF-A | 0.72 | 0.73 | 0.72 | 933 |
| | Considerate | 0.76 | 0.81 | 0.78 | 1000 | (Random Forest Model) | 0.74 | 0.82 | 0.78 | 962 |
| | High+ Very High | 0.51 | 0.13 | 0.21 | 126 | | 0.42 | 0.09 | 0.15 | 130 |
| Accuracy =0.76 ||||| 2920 | Accuracy =0.76 |||| 2920 |
| | | | | | | | | | | |
| (c) | Low | 0.93 | 0.78 | 0.85 | 1400 | (d) | 0.87 | 0.90 | 0.88 | 1400 |
| RF-1 | Medium | 0.67 | 0.70 | 0.68 | 1316 | RF-2 | 0.73 | 0.67 | 0.70 | 1316 |
| (Pérez-Guillén et. al., 2022) | Considerate | 0.73 | 0.84 | 0.78 | 1223 | (Pérez-Guillén et. al., 2022) | 0.76 | 0.78 | 0.77 | 1223 |
| | High + Very High | 0.64 | 0.65 | 0.64 | 133 | | 0.56 | 0.71 | 0.63 | 133 |
| Accuracy =0.76 ||||| 4072 | Accuracy =0.78 |||| 4072 |


## Appendix A (Formulas for the Evaluation metrics):

Performance indicators like accuracy, precision, Recall, and F1-score are used for assessing the effectiveness of the avalanche prediction model. The notations used are





● TP: True Positive: Number of Points that are positive and predicted to be positive

● FN: False Negative: Number of Points that are positive but predicted to be negative

● FP: False Positive: Number of Points that are negative but predicted to be positive

● TN: True Negative: Number of Points that are negative and predicted to be negative

Accuracy of classification is the ratio of correct predictions to the total number of input samples.

$$Accuracy = \frac{Number\ of\ predicitons}{Total\ number\ of\ predictions} \quad (1)$$

Precision is the total number of successfully classified positive classes to the total number of anticipated positive classes.


$$Precision = \frac{TP}{TP+FP} \quad (2)$$

A recall is the total number of correctly classified positive classes to the total number of positive classes.

$$Recall = \frac{TP}{TP+FN} \quad (3)$$

The F1 Score is the harmonic mean of precision and Recall. Mathematically, it can be expressed as

$$F1 = 2 * \frac{1}{\frac{1}{Precision}+\frac{1}{Recall}} \quad (4)$$


The formulas used for calculating the macro and weighted average are as follows.

$$weighted\ average = \frac{\sum_{i=1}^{5} w_i x_i}{\sum_{i=1}^{5} w_i} \quad (5)$$

Where $w_i$ denotes the weights of the five classes and $x_i$ denotes the value.

$$Macro\ average = \frac{\sum_{i=1}^{5} x_i}{5} \quad (6)$$

Where $x_i$ denotes the value, and 5 is the number of the target variables



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
