# Peer review of "Neural Network Model for Automated Prediction of Avalanche Danger Level"

_EGUsphere, 2023_

## Referee Comment (RC2)

**Summary of the review of the manuscript**

**The following main points highlight the strength of the paper:**

*1)* Authors have proposed a neural networkmodel for predicting avalanches, a new approach on the available good quality of data.

*2)* A mathematical model based on past data record is trained with a quality-controlled sub-dataset of Swiss Alps to predict the avalanches danger level.

*3)* The model Training accuracy of 79.75% and validation accuracy of 76.54% have been achieved, which is quite significant.

**Introduce the problem clearly and well written in structured manner:**

*1)* Accurate prediction of snow avalanches can help ensure people's safety in snow-covered regions.

*2)* The major hurdle in developing machine learning models is the lack of sufficient and reliable data. This issue has been resolvedto a great extent by the WSL Institute of Snow and Avalanche Research, Switzerland, by collecting 20 years of data inavalanche forecasting.

*3)* The data set has been further refined with data selection techniques.

*4)* The dataset combines differentfeature sets with meteorological variables.

*5)* This unique dataset has enabled experimentation with machine learning models like neural networks and compared its performance with the random forest machine learning technique.

**Analyses of the data have been done for prediction purpose:**

The dataset used for the training ofneural networks is described well. Authors explained the neural network model, tuning of hyper parameters and evaluation metrics. Random Forest machine learning method details applied to the same dataset are described.

**Relevant papers are cited.** Following two more references may be added in the paper which are using different techniques for snow avalanche.

*1)* Amreek Singh, AshwagoshaGanju, 2008. Artificial Neural Networks for Snow Avalanche Forecasting inIndian Himalaya. "The 12th International Conference ofInternational Association for Computer Methods and Advances in Geomechanics (IACMAG)1-6 October, 2008Goa, India.

*2)* Singh, A., Srinivasan, K. and Ganju, A. 2005. Avalanche Forecast Using Numerical Weather Prediction in Indian Himalaya,Cold Regions Science and Technology, Vol. 43, 83-92.

**On the basis of mentioned facts and importance of the topic I recommend this work for publication.**

---

## Author Response (AR1)

Point by point changes made in the manuscript according to the comments

Reviewer 2 :  Suggested references are included in the manuscript and are marked with violet colour.

Reviewer 3: Two recent references on use of deep learning techniques for analysing snow avalanches have been included. And explanations  regarding preprocessing techniques used are now referred to in the manuscript in Sec-3 (Dataset). All revisions are marked with red colour.